# Comparative Analysis of the nrDNA Repeat Unit of Manila Clam *Ruditapes philippinarum* and Quahog *Mercenaria mercenaria*

**Zhansheng Guo [1], Zhen Wang [2] and Xuguang Hou [1,\*]**

1 Marine College, Shandong University, Weihai 264209, China; guozhansheng@sdu.edu.cn
2 Binzhou Institute of Scientific and Technical Information, Binzhou 256600, China; bzskjjwzs@ba.shandong.cn
\* Correspondence: houxuguang@sdu.edu.cn; Tel.: +86-0631-5688303

**Abstract:** *Ruditapes philippinarum* and *Mercenaria mercenaria* are economically important bivalve species. The complete ribosomal DNA (rDNA) unit sequences of *R. philippinarum* and *M. mercenaria* , with as-sembled rDNA unit lengths of 12,910 and 12,100 bp, respectively, were obtained in this study for the first time. The rDNA unit structural organisation was similar to that in other eukaryotes, in-cluding the following elements in order: 18S rRNA-internal transcribed spacer 1 (ITS1); 5.8S rRNA-ITS2-28S rRNA-intergenic spacer (IGS) (3′ external transcribed spacer (ETS); non-transcribed spacer (NTS)-5′ ETS). The genetic differences between *R. philippinarum* and *M. mercenaria* were mainly attributable to non-coding regions (ITS1, ITS2 and IGS), especially the IGS region. The boundaries of putative 3′ ETS, NTS and 5′ ETS were confirmed. Seven and three sub-repeat fragments were found in *R. philippinarum* and *M. mercenaria* , respectively. These frag-ments ranged from 4 to 154 bp in length, and were located at the NTS and 5′ ETS regions. Five and six cytosine–guanine (CpG) islands were detected in *R. philippinarum* and *M. mercenaria* , respec-tively, and these covered 85.58% and 79.29% of the entire IGS sequence, respectively. The phylo-genetic tree was constructed based on Veneridae ITS and 18S rRNA sequences using the maxi-mum likelihood (ML) method. The ML tree based on ITS revealed that species within the same genus clearly clustered together with relatively high supporting values, and all the genera were recovered as monophyletic. The phylogenetic analyses using 18S rRNA provided a weaker phy-logenetic signal than ITS.

**Keywords:** ribosomal DNA; *Ruditapes philippinarum*; *Mercenaria mercenaria*; intergenic spacer; functional element

## 1. Introduction

Veneridae is one of the most diverse families of molluscs, with approximately 800 species [1,2]. Its members are distributed in marine environments worldwide, where they burrow in muddy or sandy habitats. Many Veneridae species are economically im-portant due to their huge abundance in benthic environments, such as the Manila clam *Ruditapes philippinarum*, the northern quahog *Mercenaria mercenaria*, the western Pacific hard clam Meretrix petechialis, the equilateral clam Gomphina aequilatera and the Taca clam Protothaca theca [1–3]. The classification system and phylogenetic re-lationships of Veneridae have been based mainly on morphological characteristics and molecular data (mitochondrial and nuclear sequences) [1,2,4–6], which are currently un-der debate. Problems with existing classification methods are attributed to (1) the con-vergence of the evolutionary and phenotypic plasticity of morphological traits in most venerids [6]; (2) dou-ble uniparental inheritance (DUI) of mitochondrial DNA in some Veneridae species [7–9]; (3) highly conserved in some nuclear markers, such as the nu-clear protein-coding gene (H3) and ribosomal 18S or 28S rRNA [5].

In eukaryotes, nuclear ribosomal DNA (nrDNA) is usually organised in long tan-dem repeats that form nucleolar organising regions in chromosomes [10,11]. In animals, each

nrDNA repeat unit has a similar structure, which consists of conserved coding genes (18S, 5.8S and 28S rRNA) separated by two internal transcribed spacers (ITS1 and ITS2) and an intergenic spacer (IGS) located between 28S rRNA and 18S rRNA [11]. The IGS region comprises two external transcribed spacers (5′ETS and 3′ETS) and a non-transcribed spacer (NTS) [11]. Each nrDNA region has evolved at a different rate and can be used as a genetic marker for phylogenetic analyses under different lev-els. Coding genes (18S and 28S rRNA) are highly conserved between organisms and are suitable for phylogenetic comparisons at the family and higher levels [12]. By con-trast, non-coding regions (ITS1, ITS2 and IGS) evolve rapidly and have high structural variability and can, thus, be used as genetic markers for phylogenetic comparisons at the genus and lower levels, including between individuals [12]. Salvi and Mariottini (2012) validated ITS2 as a suitable marker for venerid phylogenetic and taxonomic studies and showed a higher resolution below the subfamily level [5]. Despite recent studies of rDNA regions, only 11 complete ITS sequences from venerids are available in the NCBI database (Table 1). A complete rDNA unit and IGS region have never been reported.

**Table 1.** Collection information, GenBank accession numbers and the lengths of each ribosomal region for Veneridae species used in the present study.

| Subfamily | Genus | Species | ITS | | 18S | | 28S | |
|---|---|---|---|---|---|---|---|---|
| | | | GenBank Accession Number | Length (bp) | GenBank Accession Number | Length (bp) | GenBank Accession Number | Length (bp) |
| Meretricinae | *Meretrix* | *Meretrix lusoria* | DQ389108 | 1567 | JN996714 | 1899 | MW221234 | 739 |
| Meretricinae | *Meretrix* | *Meretrix meretrix* | AY695801 | 1480 | EF426291 | 1900 | / | / |
| Tapetinae | *Ruditapes* | *Ruditapes philippinarum* | DQ399404 | 1303 | JN807343 | 1833 | AM779742 | 1476 |
| Tapetinae | *Ruditapes* | *Ruditapes variegatus* | JN807381 | 1265 | JN807349 | 1832 | DQ343858 | 1388 |
| Tapetinae | *Ruditapes* | *Ruditapes decussatus* | HQ634139 | 1259 | KX713340 | 1779 | KX713427 | 2078 |
| Tapetinae | *Paratapes* | *Paphia undulata* | EU183531 | 1236 | JN996722 | 1831 | JQ277802 | 743 |
| Chioninae | *Leukoma* | *Protothaca staminea* | EF035084 | 1379 | AM774570 | 1779 | AM779744 | 1478 |
| Chioninae | *Leukoma* | *Protothaca jedoensis* | DQ220291 | 1437 | EF426292 | 1831 | DQ343856 | 1390 |
| Chioninae | *Mercenaria* | *Mercenaria mercenaria* | DQ190445 | 1332 | AF106073 | 1804 | KX713401 | 2085 |
| Callistinae | *Saxidomus* | *Saxidomus gigantea* | EF035110 | 1332 | / | / | / | / |
| Cyclininae | *Cyclina* | *Cyclina sinensis* | DQ190446 | 1263 | EF426289 | 1838 | DQ343849 | 1396 |
| | outgroup | *Perna viridis* | MH279802 | 735 | KY081342 | 1698 | MK419106 | 3679 |

The Manila clam *Ruditapes philippinarum* (Adams and Reeve, 1850) and quahog *Mercenaria mercenaria* (Linnaeus, 1758) are widespread and commercially important bivalve species. *R. philippinarum* originated in the Indo-Pacific region, but has been in-troduced to the west coast of North America and the Atlantic and Mediterranean coasts of Europe [13]. *M. mercenaria* is native to the east coast of the US and Canada and has been introduced to China for commercial cultivation [14]. Although R. philip-pinarum and *M. mercenaria* have been commercially exploited and deeply manipulated by human activities, their molecular genetics remain understudied [13,14]. The com-plete nrDNA repeat units of *R. philippinarum* and *M. mercenaria* are sequenced for the first time in this study. The two aims of this study are to analyse the structural features of each rDNA region, especially the IGS region, and to compare the phylogenetic rela-tionships with those of publicly available venerids based on ITS and 18S rRNA se-quences.

## 2. Materials and Methods

### 2.1. Sampling and DNA Extraction

Each individual of *R. philippinarum* and *M. mercenaria* was collected in May 2019 from Shuangdao Bay, Weihai, China. The adductor muscles were dissected and pre-served in 75% ethanol until further analysis. Total genomic DNA was extracted using an EasyPure® Marine Animal Genomic DNA Kit (TransGen Biotech Co., LTD., Beijing, China) according to the manufacturer's instructions.

## 2.2. PCR Amplification, Cloning and Sequencing

The nrDNA fragments were amplified using the primers listed in Table 2. The primers used to amplify the 18S rRNA gene were designed based on the conserved se-quences of the 18S rRNA genes of *R. philippinarum* (EF426293), Meretrix meretrix (EF426291) and *M. mercenaria* (JN996711). Because the lengths of the complete 28S rRNA sequences in molluscs generally exceed 3.5 kb, the 28S rRNA genes of R. philip-pinarum and *M. mercenaria* were divided into three parts for primer design to ensure successful amplification. These primers were then designed based on the 28S rRNA gene sequences of *M. mercenaria* (KX713401), Mysia undata (KX713408) and R. decus-satus (KX713427). The primers used to amplify the 18S-ITS-28S region were designed based on the 18S rRNA genes of *R. philippinarum* (EF426293) and *M. mercenaria* (JN996711) and the 28S rRNA genes of *M. mercenaria* (KX713401) and M. undata (KX713408). The 28S-IGS-18S region was designed based on the newly sequenced 5′ end of 18S and 3′ end of 28S rRNA gene sequences from *R. philippinarum* and M. mer-cenaria, respectively. For PCR amplification, each reaction had a total volume of 25 µL and contained 1.0 µL of DNA template (100 ng/µl), 0.3 µL of each primer (10 µmol/L), 12.5 µL of 2× SanTaq PCR Mix (Sangon Biotech Co., Ltd., Shanghai, China) and 10.9 µL of ddH2O. The PCR amplification protocols were as follows: initial denaturation at 95 °C for 3 min; 35 cycles of denaturation at 95 °C for 30 s; annealing at 53–59 °C for 45 s; extension at 72 °C for 1 min–6 min 30 s (Table 2). All reactions included a final exten-sion at 72 °C for 7 min.

**Table 2.** PCR amplification primers.

| Amplified Region | Primers | Sequences (5′–3′) | Annealing Temperature | Extension Time | Length (bp) |
|---|---|---|---|---|---|
| 18S rRNA | 18S-ar | TCAAATGTCTGCCCTATC | 53 °C for *R. philippinarum*, 55 °C for *M. mercenaria* | 1 min 30 s | 1501–1573 |
| | 18S-br | TTCACCTACGGATACCTTG | | | |
| 18S rRNA–ITS-28S rRNA | ITS-ar | TAACAAGGTATCCGTAGGTG | 53 °C for *R. philippinarum*, 55 °C for *M. mercenaria* | 1 min | 1436–1841 |
| | ITS-br | CGTGCCAGTATTTAGCC | | | |
| 28S rRNA | 28S1-ar | AGTCGGGTTGTTTGGGAATG | 55 °C | 1 min | 1240–1251 |
| | 28S1-br | TTGATTCGGCAGGTGAGTTG | | | |
| | 28S2-ar | CTGTGGGATGAACCAAACGC | 55 °C | 1 min 30 s | 1649–1691 |
| | 28S2-br | ACCTTAGGACACCTGCGTTA | | | |
| | 28S3-ar | TCACCCACTAATAGGGAACG | 55 °C | 1 min | 691–702 |
| | 28S3-br | AAGCACCTAAACCAAATGTC | | | |
| IGS for *R. philippinarum* | IGS-ar1 | CCAAATGCCTCGTCATCTAA | 59 °C | 6 min 30 s | 7829 |
| | IGS-br1 | CTGCCTTCCTTGGATGTG | | | |
| IGS for *M. mercenaria* | IGS-ar2 | GAATACAGACCGTGAAAGCG | 56 °C | 5 min 30 s | 6434 |
| | IGS-br1 | CTGCCTTCCTTGGATGTG | | | |

All PCR products were electrophoresed on 1.0% agarose gels and then extracted and purified using SanPrep Column DNA Gel Extraction Kit (Sangon Biotech Co., Ltd., Shanghai, China) according to the manufacturer's instructions. As the lengths of 28S-IGS-18S region of *R. philippinarum* and *M. mercenaria* both exceeded 6 kb, which were hard to cloned successfully. The purified PCR products, except for the IGS region, were cloned as described by Guo et al. (2019) [11]. Three positive colonies were picked and sequenced by Sangon Biotech Co., Ltd. (Shanghai, China). The purified IGS region was quantified using the Qubit® 2.0 Fluorometer (Invitrogen, Carlsbad, CA, USA), and submitted to Sangon Biotech (Shanghai, China) for library construction and high-throughput sequencing on the Illumina HiSeq 2500 platform.

## 2.3. Sequence Analysis

The high-throughput sequencing reads were filtered by trimming based on low quality. Cleaned reads were assembled using Geneious v7.0 (Biomatters, Auckland, New Zealand). Annotations were performed, and the IGS region was confirmed us-

ing the BLAST tool from Geneious software and the NCBI database [15]. The complete rDNA unit was assembled using DNAMAN v10.0 [16]. The boundaries of each region were confirmed according to Guo et al. (2018) [11]. The regions were submitted to the GenBank database under the following accession numbers: MZ227551-MZ227552 (18S rRNA), MZ241537-MZ241538 (ITS), MZ227553-MZ227554 (28S rRNA) and MZ274272-MZ274273 (IGS). Multiple sequence alignment was performed using MAFFT online tool, and the parameters were set as default [17]. The general molecular features of *R. philippinarum* and *M. mercenaria* rDNA were calculated using MEGA 11 [18]. Sub-repeat (SR) fragments of the IGS region were detected using Tandem Repeats Finder software (https://tandem.bu.edu/trf/trf.html, accessed on 8 June 2021) [19]. The putative transcription initiation sites (TISs) and transcription termination sites (TTSs) were predicted based on other mollusc species [11,20,21]. CpG islands were confirmed with EMBOSS CpGPlot software (https://www.ebi.ac.uk/Tools/seqstats/emboss_cpgplot/, accessed on 2 June 2021) [17]. The sequence identity was confirmed using BioEdit v7.2.

For phylogenetic reconstruction, the complete ITS sequences (ITS1-5.8S-ITS2) and 18S rRNA sequences of *R. philippinarum* and *M. mercenaria* were determined. The re-maining sequences of Veneridae species were picked up from the GenBank database (Table 1). Perna viridis (MH279802 for ITS and KY081342 for 18S rRNA) was selected as the outgroup. Maximum likelihood (ML) phylogenetic trees were constructed with MEGA 11 using pairwise distances based on the GTR+G+I and GTR+G model of nucle-otide substitution for ITS and 18S rRNA, respectively. Bootstrapping (1000 replicates) was used to gauge support for individual clades in the resulting phylogenetic tree.

## 3. Results

### 3.1. Complete nrDNA Sequences of R. philippinarum and M. mercenaria

The complete rDNA units of *R. philippinarum* and *M. mercenaria* were determined after amplification, sequencing and assembly. In agreement with data from other eukaryotes, the structural organisation of the rDNA repeat units included the following elements in order: 18S rRNA-ITS1, 5.8S rRNA-ITS2 and 28S rRNA-IGS. The lengths of the complete rDNA repeat unit were 12,910 and 12,100 bp in *R. philippinarum* and *M. mercenaria*, respectively, and the respective GC contents were 59.14% and 57.52%, respectively.

The length, GC content, pairwise identity and variable sites of each region are presented in Table 3. The GC content of each region exceeded 51% in both *R. philippinarum* and *M. mercenaria*, and this value exceeded 63% in ITS1 and ITS2. The encoding genes (18S, 5.8S and 28S rRNA) in *R. philippinarum* and *M. mercenaria* had low levels of genetic difference, in addition to pairwise identities of 99.13%, 100% and 96.65%, respectively. A total of 12 and 90 variable sites were detected within 18S rRNA and 28S rRNA, respectively. The differences between *R. philippinarum* and *M. mercenaria* were observed mainly in non-coding regions (ITS1, ITS2 and IGS). The lengths of ITS1 and ITS2 had no evident differences between *R. philippinarum* and *M. mercenaria*; however, these regions contained 254 and 133 variable sites, respectively, which could have resulted from the relatively low sequence identities (46.24% in ITS1, 61.66% in ITS2). Each region of the rDNA units was compared, and the lowest pairwise identity (40.47%) and highest size difference (6216 bp in *R. philippinarum* vs. 5402 bp in *M. mercenaria*) were found in the IGS region. The contribution of variable sites of the IGS sequences between two species reached up to 43.66%.

**Table 3.** Characterisation of rDNA from *R. philippinarum* and *M. mercenaria*.

| Region | *Ruditapes philippinarum* | | *Mercenaria mercenaria* | | Alignment Length (bp) | Pairwise Identity (%) | Pairwise Distance | Variable Site (V) |
|---|---|---|---|---|---|---|---|---|
| | Length (bp) | GC Content (%) | Length (bp) | GC Content (%) | | | | |
| 18S rRNA | 1832 | 51.85 | 1830 | 51.75 | 1833 | 99.13 | 0.007 | 12 |
| ITS1 | 638 | 64.42 | 679 | 63.04 | 692 | 46.24 | 0.599 | 254 |
| 5.8S rRNA | 157 | 57.96 | 157 | 57.96 | 157 | 100.00 | 0.000 | 0 |
| ITS2 | 434 | 65.67 | 420 | 65.72 | 446 | 61.66 | 0.432 | 133 |
| 28S rRNA | 3633 | 57.68 | 3612 | 57.22 | 3638 | 96.65 | 0.025 | 90 |
| IGS | 6216 | 61.18 | 5402 | 58.33 | 6239 | 40.47 | 0.878 | 2724 |

### 3.2. Informative Characteristics of the IGS Region

The IGS region contains several functional elements, including TTS, TIS and SR fragments and CpG islands (Figure 1). A Thymine-rich tract was identified in the IGS region of *R. philippinarum* from bp 198 to 267, and *M. mercenaria* had two ploy (T) tracts at positions bp 191–217 and 274–292. These sequences were considered as putative TTSs. Therefore, the positions of 3′ ETS in the *R. philippinarum* and *M. mercenaria* IGS regions were at bp 1–267 and 1–292, respectively. The sequences TAAGAAGGCAGGCTACGGGCGGG and GGGAGGG in *R. philippinarum* and *M. mercenaria* were considered the putative TISs for RNA polymerase I, and a nucleotide at positions 3760 bp and 3742 bp, respectively, was identified as the + 1 position of the TIS. The 5′ ETS was located between the putative TIS and 5′ 18S rRNA, and the remainder of the IGS contained the NTS. The respective sizes of the 5′ ETS and NTS were 3492 and 2457 bp in *R. philippinarum* and 3449 and 1661 bp in *M. mercenaria*.

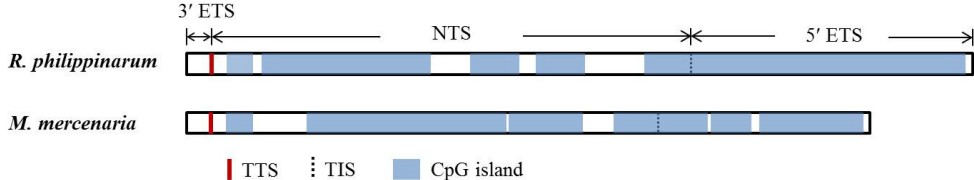

**Figure 1.** Structural organisation scheme of the IGS region in *R. philippinarum* and *M. mercenaria*. ETS, external transcribed spacer; NTS, non-transcribed spacer; TTS, transcription termination site; TIS, transcription initiation site; CpG island, cytosine-guanine island.

The DNA repeats in the IGS sequences of *R. philippinarum* and *M. mercenaria* were analysed, and the length, copy number, location, GC content and entropy of each sub-repeat fragment are listed in Table 4. *Ruditapes philippinarum* and *M. mercenaria* contained seven and three SRs, respectively, which ranged from 4 to 154 bp and were located at the NTS and 5′ ETS, respectively. The numbers and lengths of SR copies varied widely between *R. philippinarum* and *M. mercenaria*. In *R. philippinarum*, SR1 (length: 36 bp) was present in 3.8 copies; SR2 (length: 154 bp) was repeated three times; SR5 was found in only 1.9 repeats. By contrast, in *M. mercenaria*, SR1 and SR3 were both 48 bp in size and repeated three times. SR2 (96 bp) was present in 1.8 copies.

**Table 4.** Characterisation of repeated regions defined for *R. philippinarum* and *M. mercenaria*.

| Species | Sub-Repeat | Width (bp) | Copy Number | Location | GC Content (%) | Entropy (0–2) |
|---------|-----------|-----------|-------------|----------|----------------|---------------|
| *Ruditapes philippinarum* | R-S1 | 36 | 3.8 | 592–727 | 55.65 | 1.99 |
| | R-S2 | 154 | 3 | 700–1161 | 54.55 | 1.98 |
| | R-S3 | 4 | 29 | 2635–2752 | 75.00 | 1.49 |
| | R-S4 | 62 | 2 | 3164–3284 | 63.64 | 1.90 |
| | R-S5 | 72 | 1.9 | 3320–3454 | 65.75 | 1.89 |
| | R-S6 | 141 | 2 | 4338–4619 | 65.96 | 1.91 |
| | R-S7 | 72 | 1.9 | 4637–4771 | 65.75 | 1.89 |
| *Mercenaria mercenaria* | M-S1 | 48 | 3 | 3615–3761 | 66.67 | 1.89 |
| | M-S2 | 96 | 1.8 | 3599–3776 | 64.29 | 1.88 |
| | M-S3 | 48 | 3 | 4054–4197 | 62.50 | 1.91 |

The nucleotide compositions of the IGS region had a GC bias, and the GC content was slightly higher in *R. philippinarum* (61.18%) than in *M. mercenaria* (58.33%). Five and six CpG islands were detected in *R. philippinarum* and *M. mercenaria*, respectively, and were both located at the NTS and 5′ ETS. The CpG islands covered 85.58% and 79.29% of the entire IGS sequences, respectively (Figure 1). The CpG islands in *R. philippinarum* ranged from 206 to 2540 bp in size. In *M. mercenaria* they varied from 215 to 1585 bp in size.

### 3.3. Phylogenetic Analysis of Veneridae Based on ITS and 18S rRNA Sequence

Due to the lack of sufficient IGS and 28S rRNA data, ITS and 18S rRNA sequences were used to reveal the phylogenetic relationships among Veneridae species. Specifically, the ML method was used to construct a phylogenetic tree (Figure 2). The ML tree based on ITS revealed that species within the same genus clearly clustered together with relatively high supporting values. At the generic level, all the genera were recovered as monophyletic. While at the subfamilial level, Chioninae and Meretricinae formed a monophyletic and well supported clade. Tapetinae was paraphyletic, *P. undulata* was more closely related to *S. gigantea* than genus *Ruditapes*. The phylogenetic tree based on 18S rRNA produced similar topologies with those of ITS except the position of *R. decussatus* and *P. undulata*, but with relatively low supporting values, which was in accordance with previous molecular studies [3,22].

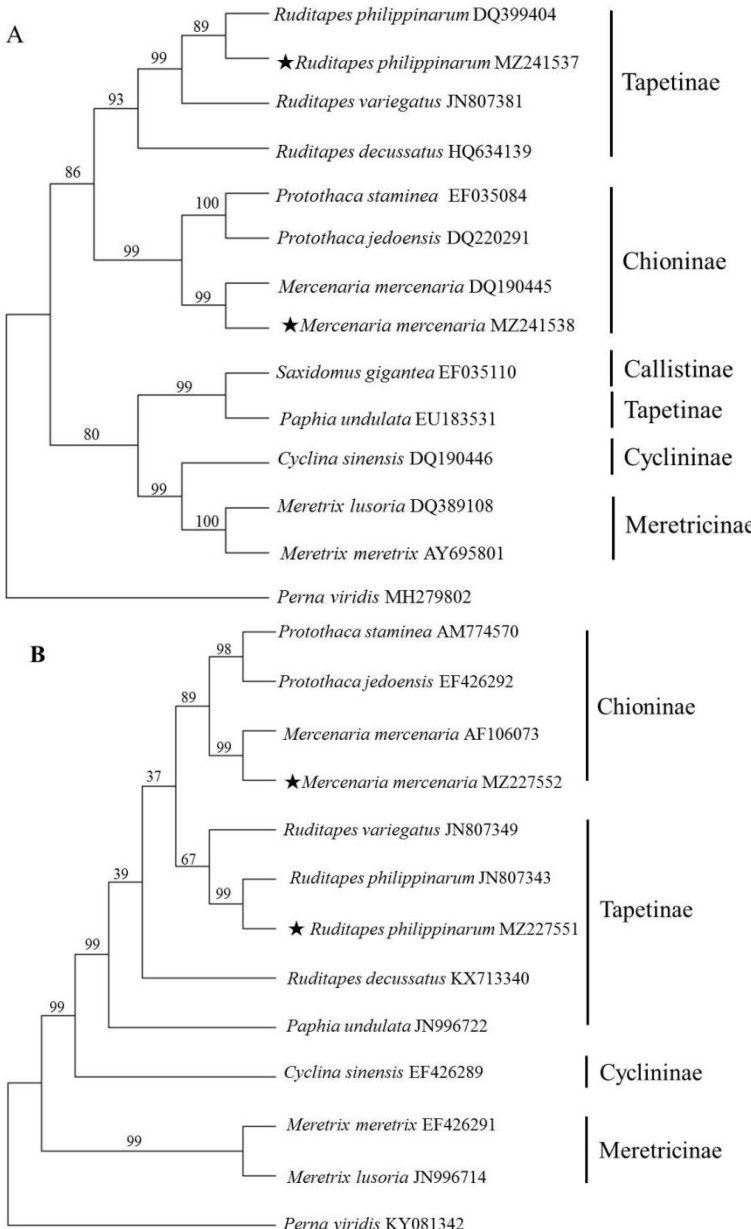

**Figure 2.** Phylogenetic trees constructed using the maximum likelihood (ML) method based on ITS (**A**) and 18S rRNA sequences (**B**) of Veneridae species. The asterisk (★) means the sample collected in the present study.

## 4. Discussion

The complete rDNA units of *R. philippinarum* and *M. mercenaria* were sequenced for the first time in this study. The structural composition and order of the rDNA units were similar to those of other reported eukaryotes [10–12,20,21,23–28]. The rDNA unit lengths of the reported marine invertebrates ranged from 7.6 to 12.91 kb [11,20,21,23–26], and *R. philippinarum* had the longest rDNA (12.91 kb) among these taxa. The rDNA units of marine invertebrates were significantly shorter than those of mammals (43 kb in human, 45 kb in mouse) [27,28]. The differences in rDNA lengths between marine invertebrates were mainly attributable to the IGS, which varied in length from 1399 bp in *Perna canaliculus* [11] to 6216 bp in *R. philippinarum*.

Different copies of rDNA usually evolve in a concerted manner within species but independently between species [29]. However, if the mutation rate of a particular rDNA sequence occurs faster than that of homogenisation during evolution, multiple variations will cause rDNA polymorphisms, namely, non-concerted or incomplete concerted evolution [29–31]. rDNA polymorphism has been reported in *Haliotis tuberculate* [32], *Pleuronichthys cornutus* [29], *Cynoglossus zanzibarensis* [30], Soleidae [31] and *Fusarium proliferatum* [33]. A comparison of the ITS1 and ITS2 sequences between samples in this study and others that were submitted to the NCBI GenBank database led to the detection of three ITS1 types in *R. philippinarum*. Type one was the identity of the whole ITS1 sequence, which exceeded 99% with no mismatch fragments, e.g., MZ241537. Type two was a mismatch fragment detected from 485 to 517 bp, e.g., AY498755. Type three was a mismatch fragment detected from 485 to 524 bp, e.g., GU358316. The identities of the other type two and three sequences exceeded 99%. Except for ITS1 in *R. philippinarum*, no second ITS1 or ITS2-type was detected in *R. philippinarum* or *M. mercenaria*.

The 11 ITS sequences used to construct the phylogenetic tree belonged to five sub-families. Species assigned to the same sub-family were placed in the same clade except for *P. undulata*, which clustered with the genus *Meretrix* with a high supporting value (99%). This finding agreed with Salvi and Mariottini (2012) [5]. The mitochondrial genome analysis showed that *P. undulata* was clustered with other Tapetinae species [3,6]. Just four partial ITS sequences of *P. undulata* were deposited in the GenBank database (JN996788, JN996787, JN996750 and EU183531), and the identity of ITS1 and ITS2 of them were just 36.16% and 58.25%, respectively. More *P. undulata* samples should be cloned and sequenced to confirm the accuracy and phylogenetic position of the sequence.

A Thymine rich tract at the beginning of the 5′ IGS region could be considered as a putative TTS, a feature commonly found in marine invertebrates [11,20,21,23–26]. The typical structure of putative TTS was also observed in *R. philippinarum* and *M. mercenaria*. The core promoter contained TATA and GGGNGGG boxes, which appeared to be a general feature of TIS, and a AT-rich sequence has been reported upstream [11], but no TATA box and AT-rich sequence were detected in the sequences from *R. philippinarum* and *M. mercenaria*. *Miscanthus sinensis*, *Sorghum bicolor*, *Fagus sylvatica* and *Quercus suber* are also missing a canonical TATA box upstream of the putative TIS sites [34–36]. Precise transcription factor interactions are key to promoter recognition; the TATA box may not be a direct binding site for transcription factors, but is probably conserved only for DNA to be easily melted [37].

SR fragments of different sizes are a functional element of the IGS. Both intra- and inter-species variations in the copy number and length were observed for each SR type, and these variations were rarely predictable and did not demonstrate expected similarities [38]. Compared with each IGS region in *R. philippinarum* and *M. mercenaria*, the alignments showed that the relative conserved regions were at the beginning of the 3′ ETS and end of the 5′ ETS sequences. SR fragments were interspersed in the NTS and upstream region of the 5′ ETS, and heterogeneity in the IGS length could be attributed to the presence of SRs. IGS SR fragments have been demonstrated to serve as rRNA gene transcription regulatory elements, particularly as enhancers that increase the rRNA transcription rate [20,39]. Due to these differences in the SR sequences (including copy number, sequence and length)

among different species, the IGS region could be a suitable genetic marker for revealing phylogenetic inter- and intra-species relationships and could be used to distinguish venerid species, as described previously in plant and other animal studies [35,37,40,41].

## 5. Conclusions

We performed, for the first time, a comparative analysis of the complete rDNA units of *R. philippinarum* and *M. mercenaria*. The structural organisation of these rDNA units was identical to that in other eukaryotes, with assembled rDNA unit lengths of 12,910 and 12,100 bp, respectively. Identified variations were mainly attributable to the non-coding regions, especially IGS. We also analysed the main features of IGS functional elements, namely, the TIS, TTS and SR and CpG islands. Our results suggest that *R. philippinarum* and *M. mercenaria* can provide a structural nuclear rDNA model for molecular comparisons, and this model may also contribute to analyses of Veneridae population genetics and evolution. Notably, the complete rDNA unit sequence was only determined from these two species, which likely limits the obtained phylogenetic information. Additional Veneridae species should be sequenced to clarify the phylogenetic relationships in this family.

**Author Contributions:** Conceptualization, Z.G.; Data curation, Z.W.; Methodology, Z.G.; Project administration, X.H.; Resources, X.H.; Software, Z.G. and Z.W.; Writing—original draft, Z.G. and Z.W.; Writing—review and editing, X.H. All authors have read and agreed to the published version of the manuscript.

**Funding:** The study was supported by the Fundamental Research Funds for the Central Universities (2019ZRJC006).

**Data Availability Statement:** MDPI Research Data Policies.

**Conflicts of Interest:** The authors declare no conflict of interest.

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
