# Peer review of "Comparative Analysis of the nrDNA Repeat Unit of Manila Clam Ruditapes philippinarum and Quahog Mercenaria mercenaria"

_fishes, doi:10.3390/fishes6030042_

Round 1

Reviewer 1 Report

Guo and colleagues performed a comparative study of the complete rDNA units of two economically important bivalve species. I found that the comparison of structural organization of rDNA units of these two species is worth to report. However, authors should more clarify the phylogenetic results. ML tree based on ITS sequences showed the relatively low level of supports in most of deep nodes. This might hardly tell us the usefulness of ITS marker to resolve this group of species. If the distinct clusters of five subfamilies were identified, it should be clearly explained and show them in the main text and in figure 2. Additionally, there is no section about how to phylogenetic analyses were performed in materials and methods. Authors should include the section for the alignment, assembly, and phylogenetic construction methods.  

Minor comments

  • Delete the first paragraph in the introduction which is information for authors.
  • Explain more details why this family is economically important (line 41).
  • Show the species name in italic and present subfamily names in the phylogeny (Fig. 2)
  • Write full name of genus at the beginning of the sentence. R. philippinarum -> Ruditapes (line 152)

Author Response

  1. (1) ML tree based on ITS sequences showed the relatively low level of supports in most of deep nodes. This might hardly tell us the usefulness of ITS marker to resolve this group of species. If the distinct clusters of five subfamilies were identified, it should be clearly explained and show them in the main text and in figure 2. (2) There is no section about how to phylogenetic analyses were performed in materials and methods.

Reply: Thank you very much for your comments. (1) The phylogenetic relationships of Veneridae have been studied based on 18S rRNA and 28S rRNA partial sequences, and the present study was attempted to analyze whether ITS was a suitable marker, although the supports in some nodes were relatively low. Ruditapes philippinarum (GeneBank accession number: JN807374) and Ruditapes variegatus (JN807381) were submitted by Kang, H.-S., Kim,B.-K. and Choi,K.-S (2012), in order to avoid the interference of geolocation and experiment, another Ruditapes philippinarum (DQ399404) were selected for phylogenetic analysis. Besides, the phylogenetic tree based on 18S rRNA was constructed for comparing that of ITS, the details were shown in the revised manuscript. (2) We are sorry for our careless mistakes, we forgot to paste the section ‘2.3 Sequence analysis’ from our original paper to the journal template file, we have added this section in the revised manuscript.  

  1. Delete the first paragraph in the introduction which is information for authors.

Reply: Thank you very much. We have deleted the first paragraph in the introduction.

  1. Explain more details why this family is economically important (line 41).

Reply: We have added more details about the economically important of Veneridae according to the reviewer’s suggestion, the details were shown in the revised manuscript.  

  1. Show the species name in italic and present subfamily names in the phylogeny (Fig. 2)

Reply: Thank you very much for your comments, we have changed the species name in italic and presented subfamily name in the revised manuscript.

  1. Write full name of genus at the beginning of the sentence. R. philippinarum -> Ruditapes (line 152)

Reply: Thank you very much. The statements of ‘R. philippinarum and M. mercenaria contained seven and three SRs’ were corrected as ‘Ruditapes philippinarum and M. mercenaria contained seven and three SRs’ in the revised manuscript.  

Reviewer 2 Report

Overall:

Authors are describing complete nrDNA for two clam species. The novel aspect of this study is description of the ribosomal intergenic spacer IGS between 28S-18S. Rest of the data is probably already (submitted by other researchers) in GeneBank in form of single records like ITS-based complete sequences in Table 1 (line 67). But GeneBank accession numbers for all of the acquired sequences in this study are missing (except maybe MZ241537 and MZ241538 ITS sequences).

Moreover, what I expected to see in this paper was a comparison how well ITS sequences work vs the 28S/18S ribosomal sequences in phylogenetic reconstructions. Phylogenetic tree in fig. 2 is in my opinion not resolved and this are not “relatively high supporting values” (line 171).

I would assume that clustering of R. philllippinarum together with R.veriegatus under the same sequence of R. philllippinarum from different species as a small (supporting values are weak) proof that ITS sequences for this species are not a very good phylogenetic markers. Also, the pairwise distance for ITS 0.59 and 0.43 are so big that they can be interpreted as random noise. Structure at some point stops being conserved and we can’t be even sure that alignment of the sequences is correct (maybe structural alignment with software like LOCARNA would be useful). Because ITS is under low evolutionary pressure there is no invisible force keeping this genetic fragment in check like in case of rRNA and protein genes.

In my opinion for bivalves the small differences in 18S and 28S ribosomal sequences are more useful and informative from phylogenetic point of view. But I’m not an expert in this field so, please prove me wrong.

Personally I find works describing long nuclear loci very useful and would like to see more phylogenetic reconstructions in bivalves based on nuclear sequences (right now most are focused on mitochondrial genes/genomes) or even whole nuclear transcriptomes but this require population of the genetics databases with properly curated nuclear data.

Introduction

Line 30-38: Should be deleted.

Methodology

Line 100-102: Reagents used in PCR are described only by volume used. Initial concentration of used DNA template,  concentration of primers is missing. It would be even better if the final concentration of reagents in PCR solution was described. In that case you don’t have to write volumes of used reagents.  

Line 106-107: “PCR products were electrophoresed on 1.0% agarose gels and then extracted and purified.” Where they cut-out form the gel and purified on columns? What gel-out kit was used to do that?

Why IGS PCR product was not cloned with the rest of the PCR products? Why where the PCR products cloned in the first place? Was it due to the variability of the sequence compositions in different copies of 28S/18S/5.8S/ITS on the genome? Was those clones sequenced with Sanger method?
Line 110-112: What was the procedure for assembly of NGS data? What software was used?

Results

Line 138: First time abbreviations like TTS, TIS, SR are used there is no explanation what they mean. The only place where this abbreviations are explained is legend for Figure 1.

Line 140: The poly dT tracks. In R.phillipinarum  there are 69 repeats of nucleotide dT. How long were the reads from the Illumina HiSeq 2500? This would be impossible to sequence reliably using Sanger method. Was the NGS library TruSeq PCR-Free?

Line 169: What program was used to reconstruct phylogenetic data, what parameters? What programs and tools were used for most of the analyzes. MEGA? How about phyl. tree in Beast2?

Discussion:

Line 201: “1,399 bp in Perna canaliculus.” Source Article? Citation.

Author Response

  1. GeneBank accession numbersfor all of the acquired sequences in this study are missing (except maybe MZ241537 and MZ241538 ITS sequences).

Reply: We are sorry for our careless mistakes, we forgot to paste the section ‘2.3 Sequence analysis’ from our original paper to the journal template file, the GeneBank accession numbers for all of the acquired sequences in this study were shown in this section, which were MZ227551 - MZ227552 (18S rRNA), MZ241537-MZ241538 (ITS), MZ227553-MZ227554 (28S rRNA) and MZ274272 - MZ274273 (IGS), respectively.  

  1. (1) what I expected to see in this paper was a comparison how well ITS sequences work vs the 28S/18S ribosomal sequences in phylogenetic reconstructions. Phylogenetic tree in fig. 2 is in my opinion not resolved and this are not “relatively high supporting values” (line 171). (2) I would assume that clustering of  philllippinarumtogether with R.veriegatus under the same sequence of R. philllippinarum from different species as a small (supporting values are weak) proof that ITS sequences for this species are not a very good phylogenetic markers. Also, the pairwise distance for ITS 0.59 and 0.43 are so big that they can be interpreted as random noise. Structure at some point stops being conserved and we can’t be even sure that alignment of the sequences is correct (maybe structural alignment with software like LOCARNA would be useful). Because ITS is under low evolutionary pressure there is no invisible force keeping this genetic fragment in check like in case of rRNA and protein genes. (3) In my opinion for bivalves the small differences in 18S and 28S ribosomal sequences are more useful and informative from phylogenetic point of view. But I’m not an expert in this field so, please prove me wrong.

Reply: Thank you very much for your comments. (1) Ruditapes philippinarum (GeneBank accession number for ITS: JN807374) and Ruditapes variegatus (JN807381) were submitted by Kang, H.-S., Kim,B.-K. and Choi,K.-S (2012), in order to avoid the interference of geolocation and experiment, another Ruditapes philippinarum sample (DQ399404) were selected for phylogenetic analysis. Due to the lack of sufficient IGS and 28S rRNA data, ITS and 18S rRNA sequences were used to reveal the phylogenetic relationships among Veneridae species. 18S rRNA provided weaker phylogenetic signal than ITS, Chen et al. (2011) and Cheng et al. (2008) also revealed that the phylogenetic analyses using 18S rRNA provided the poor structure at deep levels. (2) ITS1 and ITS2 regions in Veneridae exhibit sequence variation and obvious length polymorphisms, which might be a suitable genetic markers for phylogenetic comparisons at the genus and lower levels, the feasibility should be further assessed in the next step. (3) The current studies revealed that 18S rRNA and 28S rRNA provided the poor phylogenetic resolution, and the phenomenon was also observed in the present study.   

  1. Line 30-38: Should be deleted.

Reply: Thank you very much. We have deleted the first paragraph in the introduction.

  1. Line 100-102: Reagents used in PCR are described only by volume used. Initial concentration of used DNA template, concentration of primers is missing. It would be even better if the final concentration of reagents in PCR solution was described. In that case you don’t have to write volumes of used reagents.

Reply: Thanks very much for your comments, we have added the initial concentration of DNA template and primers according to the reviewer’s suggestion in the revised manuscript. 

  1. Line 106-107: “PCR products were electrophoresed on 1.0% agarose gels and then extracted and purified.” Where they cut-out form the gel and purified on columns? What gel-out kit was used to do that?

Reply: Thank you very much. The PCR products of all sequences in this study were purified using SanPrep Column DNA Gel Extraction Kit (Sangon Biotech Co., Ltd, Shanghai, China) according to the manufacturer’s instructions.

  1. Why IGS PCR product was not cloned with the rest of the PCR products? Why where the PCR products cloned in the first place? Was it due to the variability of the sequence compositions in different copies of 28S/18S/5.8S/ITS on the genome? Was those clones sequenced with Sanger method? 

Reply: Thank you very much for your comments. (1)As the lengths of 28S-IGS-18S region of R. philippinarum and M. mercenaria were both exceeded 6 kb, which were hardly to be cloned successfully. (2) Just as the reviewer said, nuclear ribosomal DNA is a cluster structure composed of multiple tandem transcription units, in order to identify whether the sequence have variability, the PCR products were cloned in the first place in this study. (3) These clones were sequenced using 3730XL DNA analyzer with Sanger method.    

  1. Line 110-112: What was the procedure for assembly of NGS data? What software was used?

Reply: We are sorry for our careless mistakes, we forgot to paste the section ‘2.3 Sequence analysis’ from our original paper to the journal template file, we have added this section in the revised manuscript. The high-throughput sequencing reads were filtered by trimming based on low quality. Cleaned reads were assembled using Geneious v7.0 (Biomatters, New Zealand). Annotations were performed, and the IGS region was confirmed using the BLAST tool from Geneious software and the NCBI database.

  1. Line 138: First time abbreviations like TTS, TIS, SR are used there is no explanation what they mean. The only place where this abbreviations are explained is legend for Figure 1.

Reply: Thank you very much for your comments. TTS, TIS and SR are the abbreviations of the putative transcription initiation sites (TIS), transcription termination sites (TTS) and sub-repeat (SR), respectively, and which have been presented in the section of ‘2.3 Sequence analysis’.

  1. Line 140: The poly dT tracks. In phillipinarum there are 69 repeats of nucleotide dT. How long were the reads from the Illumina HiSeq 2500? This would be impossible to sequence reliably using Sanger method. Was the NGS library TruSeq PCR-Free?

Reply: Thank you very much for your comments. (1) A poly (T) tracts ( TTACTTGTTGCTT TTTGATTTATTTCTTTTCTATGTATATGTGTGGGGGAACTTTTTTTCAATTTTATTTTTTT) was identified in R.phillipinarum with a length of 69 bp. 18S rRNA-IGS-28S rRNA region in this study was sequenced using the 2 × 150 bp Miseq chemistry on an Illumina HiSeq 2500 sequencing platform, and Sanger method was not apply for this region. (2) The NGS library was constructed using NEBNext Ultra II DNA Library Prep Kit for Illumina.

  1. Line 169: What program was used to reconstruct phylogenetic data, what parameters? What programs and tools were used for most of the analyzes. MEGA?

Reply: We are sorry for our careless mistakes, we forgot to paste the section ‘2.3 Sequence analysis’ from our original paper to the journal template file, we have added this section in the revised manuscript. For phylogenetic reconstruction, the complete ITS sequences (ITS1-5.8S-ITS2) and 18S rRNA sequences of R. philippinarum and M. mercenaria were determined. The remaining sequences of Veneridae species were picked up from the GenBank database (Table 1). Perna viridis (MH279802 for ITS and KY081342 for 18S rRNA) was selected as the outgroup. Maximum likelihood (ML) phylogenetic trees were constructed with MEGA 11 using pairwise distances based on the GTR+G+I and GTR+G model of nucleotide substitution for ITS and 18S rRNA, respectively.  Bootstrapping (1,000 replicates) was used to gauge support for individual clades in the resulting phylogenetic tree.

  1. Line 201: “1,399 bp in Perna canaliculus.” Source Article? Citation.

Reply: Thank you very much, we have added the citation in the revised manuscript.

Reference

  1. Chen, J.; Li, Q.; Kong, L.; Zheng, X. Molecular phylogeny of venus clams (Mollusca, Bivalvia, Veneridae) with emphasis on the systematic position of taxa along the coast of mainland China. Scr. 2011, 40, 260–271.
  2. Cheng, H.L., Peng, Y.X., Wang, F., Meng X.P., Yan, B.L., Dong, Z.G. Sequence analysis of 18S rRNA gene of six Veneridae clams (Mollusca: Bivalvia). Fish. Sci. Chin. 2008, 15, 559–567 (In Chinese with English abstract).

Reviewer 3 Report

"Comparative analysis of the nrDNA repeat unit of Manila clam ..." by Guo et al. details the first entire sequence of the nuclear ribosomal array for two commercially important clams. The arrangement of each array, characteristics of repeated units, and phylogenetic utility of ITS are reported. The paper is a careful and well-written study that fills a small gap in our knowledge and perhaps assists venerid phylogenetics.

I see three areas in need of revision. 

First, some proofreading and rewriting is required.

  • At the beginning of the paper, the keywords are missing, and the first paragraph of the introduction is actually the instructions to the authors.
  • The common names used here for M. mercenaria (hard clam and hard-shell clam) I know are on the internet, but I have always heard quahog as a common name; "hard clam" is a bit nonsensical, as most clams are hard, and I wouldn't encourage its use.
  • Lines 43-44 are oddly worded, since you say that venerid phylogenetics have been based "mainly" on morphology and DNA, and then for the latter, "mainly" on nuclear and mitochondrial DNA. My question is, what other characters have been used and what other DNA has been used if not the ones you mention.
  • When talking about the typical structure of the nrDNA array, its "usual" appearance in eukaryotes is described, then in animals this structure is described as "similar." In the discussion, however, the structural composition and order in these two species of clams is described as "identical  to those of other eukaryotes." This all seems quite vague, and it implies there is variation in nrDNA organization but then that they are all the same in this regard. I find it difficult to believe that ALL eukaryotes have the same organization of their array, but perhaps they do -- please confirm and/or add more detail.

Second, the phylogenetic analysis is  superficial. There is no mention of methods, the most important of which would be how the sequences were aligned and perhaps how they were trimmed (if done). There is no mention of the program, how the outgroup was chosen, how any missing data was dealt with, and how the ingroups were decided. The description of the tree and reference to such things as "clusters" suggest that the authors have never really read a phylogenetics paper or have any experience with phylogenetics. It should be stated what groups were "recovered as monophyletic" (or polyphyletic or paraphyletic)' it seems that all the genera were recovered as monophyletic, except for Ruditapes, which was polyphyletic. Of course, the support values at the deeper nodes is weak, so in many trees Ruditapes was probably monophyletic. The bootstrap supports are strangely described; you mean you did 1,000 pseudoreplicates? All this should be put into the methods.

Third, I am a bit worried that the whole paper is outdated. Why would anyone today simply use ITS to do a phylogenetic analysis when they can, with a bit of funding, do whole genomes? The same amount of funding needed to go collect enough species to do a good phylogenetic analysis could be used to fund a lot of sequencing. In short, would anyone really use this information to help them do a phylogenetic analysis today? Perhaps the family is so large (how many species?), and small-scale phylogenetic analyses so useful in this unresolved family (can we get more detail on the state of phylogenetics in this group?), the case can be made that easily sequenced ITS can be used when whole genome sequencing is beyond the reach of the lab. Also, it is reported that 18S and 28S are too conserved in the family, but does this refer to their entire sequences? Usually there are length-variable regions backed with informative characters, but only if one uses a phylogenetic method that treats indels as a character, not missing. Are they variable in the length-variable regions? Are these sections usually trimmed out before analysis? The case should really be made here for how ITS can be useful in the age of next-generation sequencing.

Author Response

  1. At the beginning of the paper, the keywords are missing, and the first paragraph of the introduction is actually the instructions to the authors.

Reply: We are sorry for our careless mistakes, we forgot to paste keywords and the section ‘2.3 Sequence analysis’ from our original paper to the journal template file, we have added the keywords and deleted the first paragraph of the introduction in the revised manuscript.

  1. The common names used here for  mercenaria(hard clam and hard-shell clam) I know are on the internet, but I have always heard quahog as a common name; "hard clam" is a bit nonsensical, as most clams are hard, and I wouldn't encourage its use.

Reply: Thank you very much for your comments. The common name of M. mercenaria ‘hard- shell clam’ has been changed to ‘quahog’ in the revised manuscript.

  1. Lines 43-44 are oddly worded, since you say that venerid phylogenetics have been based "mainly" on morphology and DNA, and then for the latter, "mainly" on nuclear and mitochondrial DNA. My question is, what other characters have been used and what other DNA has been used if not the ones you mention.

Reply: We are sorry for our confusing words. The current research on venerid phylogenetics were based on nuclear and mitochondrial DNA sequences, the statements of ‘The classification system and phylogenetic relationships of Veneridae have been based mainly on morphological characteristics and molecular data (mainly mitochondrial and nuclear sequences)’ were corrected as ‘The classification system and phylogenetic relationships of Veneridae have been based mainly on morphological characteristics and molecular data (mitochondrial and nuclear sequences)’.

  1. When talking about the typical structure of the nrDNA array, its "usual" appearance in eukaryotes is described, then in animals this structure is described as "similar." In the discussion, however, the structural composition and order in these two species of clams is described as "identical to those of other eukaryotes." This all seems quite vague, and it implies there is variation in nrDNA organization but then that they are all the same in this regard. I find it difficult to believe that ALL eukaryotes have the same organization of their array, but perhaps they do -- please confirm and/or add more detail.

Reply: Thank you very much for your comments. (1) We are sorry for our vague words. The statements of ‘The structural composition and order of the rDNA units were identical to those of other eukaryotes’ were corrected as ‘The structural composition and order of the rDNA units were similar to those of other reported eukaryotes [9-11, 18-25]’ in the discussion. (2) Until now, the complete ribosomal unit sequences have been reported in mammals (human, mouse), chicken, parasites, ant, molluscs, jellyfish, land plants and macroalgae, the structure composition and order of rDNA unit were similar, including 18S rRNA–internal transcribed spacer (ITS) 1–5.8S rRNA–ITS2–28S rRNA–intergenic spacer (IGS). While rDNA polymorphism has been identified in some eukaryotes, such as Pleuronichthys cornutus and Cynoglossus zanzibarensis.       

  1. (1) There is no mention of methods, the most important of which would be how the sequences were aligned and perhaps how they were trimmed (if done). There is no mention of the program, how the outgroup was chosen, how any missing data was dealt with, and how the ingroups were decided. (2) It should be stated what groups were "recovered as monophyletic" (or polyphyletic or paraphyletic)' it seems that all the genera were recovered as monophyletic, except for Ruditapes, which was polyphyletic. Of course, the support values at the deeper nodes is weak, so in many trees Ruditapeswas probably monophyletic. (3) The bootstrap supports are strangely described; you mean you did 1,000 pseudo replicates? All this should be put into the methods.

Reply: Thank you very much for your comments. (1) We are sorry for our careless mistakes, we forgot to paste the section ‘2.3 Sequence analysis’ from our original paper to the journal template file, phylogenetic analyses were presented in this section, the details could be seen in the revised manuscript. (2) We have rewritten this part according to the reviewer’s suggestion, the details could be seen in the revised manuscript. (3) Numbers around the branches indicate bootstrap support from 1,000 replicates, and we have put this sentence into the methods.

  1. (1) Why would anyone today simply use ITS to do a phylogenetic analysis when they can, with a bit of funding, do whole genomes? In short, would anyone really use this information to help them do a phylogenetic analysis today? Perhaps the family is so large (how many species?), and small-scale phylogenetic analyses so useful in this unresolved family (can we get more detail on the state of phylogenetics in this group?), the case can be made that easily sequenced ITS can be used when whole genome sequencing is beyond the reach of the lab. (2) It is reported that 18S and 28S are too conserved in the family, but does this refer to their entire sequences? Usually there are length-variable regions backed with informative characters, but only if one uses a phylogenetic method that treats indels as a character, not missing. Are they variable in the length-variable regions? Are these sections usually trimmed out before analysis?

Reply: Thank you very much for your comments. (1) The phylogenetic relationships of Veneridae have been studied based on 18S rRNA and 28S rRNA partial sequences, and the present study was attempted to analyze whether ITS was a suitable marker, although the supports in some nodes were relatively low. Besides, the phylogenetic tree based on 18S rRNA was constructed for comparing that of ITS, the results showed that 18S rRNA provided weaker phylogenetic signal than ITS. Veneridae is one of the most diverse families of molluscs, with approximately 800 species. ITS1 and ITS2 regions in Veneridae exhibit sequence variation and obvious length polymorphisms, which might be a suitable genetic markers for phylogenetic comparisons at the genus and lower levels, the feasibility should be further assessed in the next step. (2) Although the coding regions (18S and 28S rRNA) are more higher conserved than ITS and IGS in generally, 28S rRNA has 12 divergent domains in eukaryotes (Hassouna et al., Nucleic Acids Reseach, 1984; Ellis et al., Nucleic Acids Reseach, 1986), and which have been used to analyze the phylogenetic relationships in some animals and plants. In the present study, the length variable region in 18S and 28S rRNA were not observed. In my opinion, I think that these sections should be trimmed out before analysis.       

Round 2

Reviewer 1 Report

The revised version of the manuscript reflects the reviewer's comments about phylogenetic analysis. Check below minor errors before final submission.

line 24, 193: Species —> species 

line 114: DNAMAN v10.0 needs a reference. 

Author Response

  1. line 24, 193: Species —> species 

Reply: Thanks very much for your comments. In line 24 and 193, the word “Species” has been changed to “species” in the revised manuscript.

  1. line 114: DNAMAN v10.0 needs a reference. 

Reply: Thanks very much for your comments. We have added a reference for DNAMAN according to the reviewer’s suggestion.

Reviewer 2 Report

I tried to replicate your results and I have to say making alignment of ITS-5.8S-ITS sequences is rather challenging. Please provide in article information if parameters used in MAFFT software were left as default or/and submit ITS-5.8S-ITS alignment as supplementary data.

The outgroup (P. viridis) used by authors seems to be reducing amount of data that actually is used in phylogenetic reconstruction (columns of nucleotides present in all of the aligned sequences). This is especially visible in ITS1 part (so overall most of the phylogenetic signal comes from 5.8S and ITS2 sequence fragments). In the future maybe it is worth to use software like Beast2 which do not need an outgroup sequence.

Information of Poly T tract mentioned in the article was highly misleading. I was expecting 69 repeats of nucleotide thymine. Reconsider using other name like Thymine rich tract. Line 163 and 264.

Abstract:  Last line in abstract is in different font size.

Introduction Line 60: after “quahog” there is a comma if it was left there by mistake remove it.

Material and Methods:

Line 95: Please add producent of the PCR kit. Sangon Biotech Co., Ltd, 100 Shanghai, China?

Line 119: (https://www.ebi.ac.uk/Tools/ msa/mafft/) there is a space in the address which makes it point to the wrong website …Tools/[space]msa/… please remove it. You can also remove the whole website link and add information that alignment has been made with MAFFT online tool (not downloadable software).

Literature:

Line 352-354: Citation no. 20 breaks into the next line to soon in the wrong place

Line 377-378: Citation no. 32 “Miscanthus sinensis var. glaber” the glaber part should also be italicised.

Author Response

Reviewer 2

  1. Please provide in article information if parameters used in MAFFT software were left as default or/and submit ITS-5.8S-ITS alignment as supplementary data.

Reply: Thanks very much for your comments. We have added a sentence ‘and the parameters was set as default’ in line 119 according to the reviewer’s suggestion.

  1. Information of Poly T tract mentioned in the article was highly misleading. I was expecting 69 repeats of nucleotide thymine. Reconsider using other name like Thymine rich tract. Line 163 and 264.

Reply: Thanks very much for your comments. The words ‘Poly T tract’ in line 163 and 264 were corrected as ‘Thymine rich tract’.

  1. Abstract:  Last line in abstract is in different font size.

Reply: Thank you for your careful work. We have rechecked and corrected the font of the text in the revised manuscript.

  1. Introduction Line 60: after “quahog” there is a comma if it was left there by mistake remove it.

Reply: We have removed the comma after “quahog” in Line 60 according to the reviewer’s suggestion.

  1. Line 95: Please add producent of the PCR kit. Sangon Biotech Co., Ltd, 100 Shanghai, China?

Reply: Thanks very much. We have added the producent of the PCR kit in the revised manscript.

  1. Line 119: (https://www.ebi.ac.uk/Tools/ msa/mafft/) there is a space in the address which makes it point to the wrong website …Tools/[space]msa/… please remove it. You can also remove the whole website link and add information that alignment has been made with MAFFT online tool (not downloadable software).

Reply: Thanks very much for your comments. We have removed the whole website link of MAFFT software according to the reviewer’s suggestion.

  1. Line 352-354: Citation no. 20 breaks into the next line to soon in the wrong place

Reply: Thank you very much. We have adjusted the layout of the citation no. 20 in the revised manuscript.

  1. Line 377-378: Citation no. 32 “Miscanthus sinensis  glaber” the glaberpart should also be italicised.

Reply: Thank you for your careful work. We have rechecked and corrected the font of the text in the revised manuscript.